# Public Awareness and Risk Perceptions of Endocrine Disrupting Chemicals: A Qualitative Study

**DOI:** 10.3390/ijerph17217778

**Published:** 2020-10-23

**Authors:** Melissa Kelly, Lisa Connolly, Moira Dean

**Affiliations:** Institute for Global Food Security, School of Biological Sciences, Queen’s University Belfast, BT9 5DL Belfast, Northern Ireland; melissa.kelly@qub.ac.uk (M.K.); l.connolly@qub.ac.uk (L.C.)

**Keywords:** risk perception, public perception, endocrine disruptors

## Abstract

Endocrine disrupting chemicals (EDCs) are exogenous chemicals found in food, consumer products, and the environment. EDCs are ubiquitous in modern life and exposure is associated with many negative health effects, such as reproductive disorders, metabolic disorders, and cancer. Scientists have deemed EDCs as a serious public health risk, yet the public’s perceptions of these chemicals is poorly understood. This study aimed to qualitatively explore how aware the public is of EDCs and their attitudes, beliefs, and perceptions of EDC risk. Thirty-four participants (aged 19–65 years) took part in the six focus groups. Discussions were transcribed verbatim and Nvivo 11 was used for thematic analysis. Our results indicated that awareness of EDCs was low. Themes of EDC risk perception included perceived control, perceived severity, and similarity heuristics. Risk alleviation strategies were also discussed. Future research should use quantitative methodology and a larger sample size to validate the findings from this study. Findings from this study may aid the development of effective risk communication strategies and public health interventions.

## 1. Introduction

The average person is ubiquitously exposed to low doses of endocrine disrupting chemicals (EDCs) throughout their lifetime [1]. EDCs are a group of exogenous chemicals that have the potential to significantly interfere with the endocrine functioning of animals and humans by imitating or blocking the target receptors of naturally occurring hormones in the body [2]. According to The Endocrine Disruption Exchange (TEDX), there are more than 1400 potential EDCs in the environment and food chain [3]. Examples of known EDCs include: chlorpyrifos and other pesticides; bisphenol A (BPA) and phthalates used in plastic consumer products and food containers; phytoestrogens and mycotoxins naturally found in food; lead and other heavy metals; and chemicals used in flame retardants for furniture and floor protection. Currently, EDC concentrations are regulated and monitored in accordance with government guidelines [4]. However, due to their prevalence, an increasing amount of research has been dedicated to understanding the scientific mechanisms of EDCs and their effects on human health [5].

In 2009, the Endocrine Society issued a scientific statement indicating that there was sufficient evidence to conclude that EDCs are a serious public health risk [6]. Epidemiological studies in humans have found links between EDCs and reproductive health, neurobehavioral and neurodevelopmental changes, metabolic disorders, immune disorders, and hormone related cancers [7]. As well as having associations with negative health effects, there are additional factors that contribute to the public health challenge brought on by EDCs. Despite being banned or restricted, many EDCs are resistant to environmental degradation, leading them to be potentially hazardous over an extended period of time [8]. Although EDCs are present at low-level concentrations, many can gradually accumulate in ecosystems and the tissues of organisms and can become more harmful when combined by creating a “cocktail effect” [9]. It is important to consider how low doses of a mixture of EDCs bioaccumulate and contribute to negative health conditions [9,10]. Most people are exposed to EDCs at low doses, in mixtures, and at different stages throughout their lifetime [11], with exposure beginning as early as prenatal development [12]. Finally, transgenerational effects have also been observed, for example the effects of EDCs on male fertility have shown to be biologically transferred from mother to child [13].

The complex and widespread nature of EDCs presents an obstacle for communicating the risks to the general public. Before developing effective risk communication strategies, it is imperative to first understand the public’s awareness of EDCs, and the factors that influence their risk perception [14]. Studying risk perception is an integral step in creating public health interventions, as non-contextualized information dissemination of public health risks is potentially dangerous and counterproductive [15].

Previous studies have concentrated on examining perceptions of EDCs with regard to specific demographic groups or health effects. These studies have examined awareness and risk perceptions within particular contexts, such as EDCs in water [16], EDCs and male infertility [17], and pregnant women’s perceptions of EDCs [18,19]. Results from these studies found that knowledge and awareness of EDCs was minimal. For example, Rouillon and colleagues [18,19] examined pregnant women’s knowledge, attitudes, and behaviors towards EDC exposure, and subsequently the determinants of their risk perception. Results indicated that the women’s EDC risk perception was intermediate, and their perceived severity of exposure was higher than their perceived susceptibility [18]. These women did not believe that they were particularly susceptible to exposure, but believed that exposure to EDCs was extremely dangerous. Significant determinants of their risk perception were age and level of knowledge [19].

Ho and Watanabe [20] examined the roles that perceived uncertainty and knowledge type (general vs. specific) played in EDC risk perception. The findings from this study revealed that knowledge type played significant roles in explaining risk perception, risk acceptability, and self-protective response. They found evidence that perceptions of uncertainty surrounding EDCs mediated the relationship between information type and risk perception [20]. Perceptions specifically about the controversial link between exposure to EDCs and a decline in human male fertility was examined in a study by Maxim and colleagues [17]. In this study, focus groups were conducted to gain insights into peoples’ attitudes following uncertainty communication of the effects of EDCs on male fertility. The results from these studies contradicted previous assumptions that being transparent with the general public about scientific uncertainty of EDCs elicits negative psychological effects.

Although studies on pregnant women’s perceptions and studies using the exemplar of declining male fertility offer valuable contributions to the literature regarding how different groups of people perceive EDCs, there has yet to be a study addressing how the general public perceives EDCs. There are no current studies that view EDCs as an all-encompassing group of chemicals, without targeting specific population groups or focusing on specific adverse health outcomes. Previous studies offer isolated viewpoints, and there is a need to better understand how the general public perceives EDCs as a full and contextualized picture, including associated health effects, chemical groups, and sources. Understanding the public’s perception of EDC risk is essential in further developing effective risk communication approaches.

With regard to complex public health issues, such as EDC exposure, there is an imperative need to understand the factors that shape risk perceptions. The factors and conditions that influence the EDC risk perception of the general public have yet to be explored. To address this gap, the current study set out to use a qualitative methodology to gain a better understanding of the general public’s awareness of EDCs and the specific factors that influence their risk perception. Therefore, the first objective of this study was to explore the knowledge and awareness that the general public have about EDCs, their sources, and associated health effects. The second objective was to elicit the factors that influenced public perceptions of EDC risks.

## 2. Materials and Methods

### 2.1. Study Design and Ethics

A variety of methods are available to conduct research in public health, and qualitative studies (i.e., interviews and focus groups) are better able to provide a deeper understanding of how people’s views might differ, the attitudes and beliefs held, and the factors influencing specific perspectives [21]. Focus groups generally consist of 5–10 participants and use the interactions between group members to encourage the exploration of personally held beliefs [22]. Focus groups also offer the potential to better examine how personal views are articulated and how they intersect with publicly held values, beliefs and attitudes [23]. Therefore, a qualitative descriptive study was conducted using focus groups to explore the general public’s awareness and risk perceptions of EDCs, their sources, and associated health effects. Focus groups were chosen as the mode of data collection over alternatives, such as individual interviews, as they encourage group discussions that provide clearer insights into participants’ beliefs and experiences [24]. Focus groups are also useful for describing, interpreting, contextualizing, and gaining in-depth information about complex topics [25]. During focus groups, participants may also be influenced by other group members’ comments, encouraging them to respond with their own ideas in their own words. Additionally, conducting focus groups is more time efficient, producing data from larger numbers of participants faster than individual interviews.

Ethical approval for the study was obtained by the School of Medicine, Dentistry and Biomedical Sciences Faculty Research Ethics Committee, Queen’s University Belfast. After being briefed on the study, participants gave consent to participate by signing a form that confirmed voluntary participation, confidentiality and data protection. All participants were provided with written informed consent and all procedures were approved by the School of Biological Sciences and conducted in accordance to the guidelines given in the Declaration of Helsinki.

### 2.2. Participant Recruitment

Male and female participants were recruited via convenience sampling in Northern Ireland to purposefully take part in a focus group about their attitudes towards chemicals. Recruitment emails were sent to all students and staff (technical, administrative, professional, etc.) at Queens University Belfast. Facilitator contacts and face-to-face invitations at public outreach events were also used to recruit participants. All participants were recruited using the following criteria to ensure that a diverse range of perspectives were included within the sample. Eligible participants: were aged 18–65 years; resided in Northern Ireland; were not currently pregnant; and had no prior formal education with regard to chemicals from food and/or the environment. Groups were segregated based on gender due to differences during the discussions of topics, such as fertility and reproduction. In total, 34 participants (*n* = 13 males; *n* = 21 females) were recruited for six focus groups (See Table 1). The data from the first four groups were analyzed and it was concluded that data saturation had occurred. Data saturation is a term used in qualitative research to describe when observation and analysis are no longer revealing any new information or themes [26]. Since no new themes were arising after the first four focus groups, participants in the final two focus groups were recruited from a cancer support group. These participants were a mixture of cancer patients and family members. Similar themes were being identified for this group of participants, and therefore the researchers determined data saturation had also occurred. There were no thematic differences between the groups recruited by the university and the groups recruited via the cancer support group. To maximize participation in the study, participants were given either pizza or sandwiches as a token of appreciation for their time and contribution.

### 2.3. Data Collection

Data collection took place at locations that were familiar and convenient for the participants. Data were collected through focus groups consisting of open-ended questions provided by the facilitator that encouraged discussion within the group, rather than individual responses. The focus group questions were based upon the literature and findings from previous studies on public perceptions of chemicals, chemicals mixtures, and specific EDCs. The three main topic areas were designed to elicit participants’ general and specific knowledge and awareness of EDCs, and their risk perceptions of EDCs.

The focus group topic guide (Table 2) was piloted with a mixed gender sample of the target group (*n* = 5), and further developed by the research team to assess the clarity of questions and flow from one topic to another. A postdoctoral researcher was present during the pilot group and confirmed that the conduction of the group was satisfactory. The facilitator encouraged equal participant contribution and probed when the participants did not elaborate enough on a topic. When the conversation deviated from the guide, the facilitator identified whether or not this was relevant and when necessary redirected the discussion back to the guide.

Participants were given an information packet (Appendix A) that they referred to when instructed by the facilitator at various points throughout the focus group. This was necessary as previous studies indicated that the general public are not highly aware of EDCs. Participants’ responses were based on their own knowledge. The participant packet contained information about specific EDCs and their sources, including a diagram of the human endocrine system, and was used after participants’ responses to help stimulate conversation in cases where they were unfamiliar with the topics.

Six focus group discussions took place with an average duration of 46 min. All focus groups (*n* = 6) were audio recorded with two digital recorders and took place between October 2019 and February 2020. Participants were anonymized and given a unique study ID, consisting of the number of the focus group and their participant number. The focus groups were audio recorded after verbal and written consent was obtained from the participants. Before commencing with the topic guide, the facilitator gave a brief introduction reminding the participants that their audio was being recorded, that they needed to speak clearly and that their input would remain confidential. Demographic information was also collected via handouts that also allowed participants to remain anonymous. Following the completion of the focus groups, participants were debriefed on the purpose of the focus group and given references if they wished to seek out additional information on EDCs. The facilitator also corrected any major misconceptions or inaccurate assumptions that participants had made during the focus groups.

### 2.4. Data Analysis

All recordings were transcribed verbatim by a professional transcription company and were proof read by the facilitator for accuracy. Transcripts were imported and processed in NVivo 12 Pro software (QSR International Pty Ltd., Doncaster, Victoria, Australia). The procedure of thematic analysis [27] was followed to identify themes within the data. Accordingly, transcripts were read meticulously multiple times so that familiarity with the data could be achieved. The content of the full data set was coded without using a pre-existing coding frame. A peer-review process was then used throughout the data analysis. Two researchers independently coded the same randomly selected transcripts (*n* = 2) and discussed the codes to verify the reliability and validity of their application to the data. They compared their coding which demonstrated 90% similarity (interrater reliability); differences were discussed and resolved. The remaining transcripts were independently coded and grouped into themes appearing across all six groups. The research team agreed that data saturation had occurred as no new codes emerged from the final two focus group transcripts. Participant demographic characteristics were analyzed using IBM SPSS Statistics for Windows, Version 22.0 [28].

## 3. Results

### 3.1. Participation and Sample Characteristics

Six focus group discussions were conducted in Northern Ireland. Following the completion of the fourth focus group, data saturation was achieved with no new information emerging. Thus, data collection stopped after two final focus groups.

Thirty-four participants (Table 3) took part in this study. Even though the call for participation was open to both male and female participants, mostly women came forward. Twenty-one women (aged 19–65 years) and thirteen men (aged 25–65 years) took part in the discussions with an average of six participants per group (range: three to eight). Most participants (44%) had completed an undergraduate degree and were not parents (68%). A summary of the 34 participants’ demographic characteristics is presented in Table 2.

### 3.2. Awareness of EDCs

The majority of participants were unaware of EDCs and had never heard of them before. Although it was explicit during recruitment that participants were not to have any formal education on chemicals in food and the environment, one participant revealed midway through a focus group discussion that they had previously studied food science. They claimed to recognize the phrase “endocrine disrupting chemical” but did not have any further knowledge about the chemicals. Participants recruited from the cancer support group had seen the leader of the group post about EDCs on social media, but that was the extent of their familiarity. There was one female participant who was familiar with EDCs because she was personally affected by a hormone-related disorder:
“They disrupt your hormones. They’re like oestrogen type things. I’m actually affected by it. I have Polycystic Ovarian Syndrome.”(FG2F)

Although the participants were not previously aware of EDCs, due to their knowledge of the endocrine system they were able to deduce from the name that EDCs were chemicals that interfered with the processes of this system, thus affecting the body’s hormones. Although awareness of EDCs as a general category was low, participants were more familiar with the specific EDCs. The majority of participants were familiar with pesticides and BPA due to their environmental impacts or carcinogenic properties.

Participants were aware of pesticides due to the organic movement and their negative reputation within the food industry. Additionally, three participants in FG6F were familiar with pesticides because they grew up in rural areas and recalled hearing about them from family members who were either farmers or worked with pesticides. These memories usually included the participant visualizing their family member in personal protective equipment.

There was increased familiarity with BPA due its presence in plastics. Participants had seen or used BPA-free products in the past and believed BPA to be “bad” in some way. Participants in FG2F were aware of BPA in baby bottles:
“I think there was a drive to make all babies’ bottles to BPA free, and there’s a lot of plastics that say on them BPA free, for that reason. I suppose they all contained it in the past.”(FG2F)

There was some familiarity with phthalates, dioxins, and mycotoxins. However, the information participants had about these chemicals was incomplete and there were frequent misunderstandings and confusion about these chemicals and where they are found:
“Dioxin, wasn’t it a pesticide? Years ago, it was known for causing a lot of damage to soil substrates.”(FG3M)

Overall, participants were least familiar with brominated flame retardants and phytoestrogen.

When recalling the different sources of EDCs, participants most commonly referred to plastics, food, and air particles. Participants were able to deduce that a person could be exposed to EDCs in the form of dermal absorption, inhalation, and food consumption:
“It’s from what you eat, what products you put on your skin and stuff. And you can breathe it in, it can go in through your nose and stuff like that.”(FG1F)

However, participants were not previously aware that EDCs could be biologically transferred from mother to child through breast milk and the placenta.

Birth defects and cancer were health effects that participants most commonly believed to be associated with EDC exposure. More specifically, they were aware of the relationship between the chemicals leeching from plastics increasing the likelihood of cancer.
“The plastics… it can seep into your food and cause cancer.”(FG7F)

Farmers, pregnant women, and teenagers going through puberty were most commonly believed to be the most at risk to EDC exposure. Farmers were judged to be at risk due to their proximity to harmful chemicals and their reported lack of PPE over the years. Participants believed that people may be more at risk to EDC exposure during significant periods of hormonal changes, despite there being no scientific evidence to support this. Participants identified pregnant women and teenagers going through puberty as being the most vulnerable to EDC exposure, due to their incorrect belief that more hormonal changes occur during these life stages.

### 3.3. Risk Perception of EDCs

Five major themes of EDC risk perception were identified: (1) similarity heuristics; (2) perceived severity; (3) perceived control; (4) concern for pre-existing health conditions; and (5) concern for future generations

#### 3.3.1. Similarity Heuristics

Throughout the discussions, the concept of EDCs was noted as being somewhat ambiguous and participants often used metaphors and analogies to help describe their feelings. Since EDCs were an unfamiliar topic for the majority of participants, they often used similarity heuristics to access their perceptions of EDC risk. Similarity heuristics are mental shortcuts that allow quick judgements to be made based on experiences with similar problems [26]. Participants often judged the probability of EDC risk based on its similarity to other risks with comparable long-term health outcomes, such as cholesterol and smoking cigarettes. For example, a comparison was made between the cumulative effects of EDC exposure and the development of high cholesterol to help participants make judgements about how much of EDC exposure could be genetically predisposed:
“You can eat very healthy but still have very high cholesterol. It’s hard to know then what is genetic predisposition rather than what we put into our bodies.”(FG3M)

Using these metaphors, participants often eluded to their perceptions of uncertainty surrounding the development of cancer and other health effects. They told personal anecdotes about friends or family members who had “eaten healthy, never smoked, never drank” and ended up getting cancer anyway despite their healthy choices. As mentioned previously, many participants were familiar with the movement to remove BPA from plastic products and shared that they had used BPA-free products in the past. Participants were shocked to find out that the BPA alternatives may be just as potentially harmful as BPA. In this case, the similarity heuristic was used when discussing BPA-free labelling policies, where participants compared the extraction of BPA from plastics to the removal of salt in foods.
“They’re telling you it’s BPA-free, but they’re not telling you its replacements. It’s like people taking salt out of things, but they put in mono sodium.”(FG6F)

#### 3.3.2. Perceived Severity of EDCs

All participants identified EDCs to be a far-reaching health problem that had potentially negative future consequences. The perceived severity of EDCs varied among participants. All participants acknowledged that EDCs were a serious health threat, but there were arguments that more visibly immediate threats were of higher importance. Participants believed that the general public would not classify EDCs as a high risk compared to more “important” issues, such as Brexit, getting Northern Ireland back into government, the climate crisis, and the obesity epidemic. Due to the long-term, accumulative effects that EDCs have on the body, participants recognized that EDCs were not “in the front of your mind all the time”, the way more acutely threatening risks would be:
“You don’t think about it constantly. The decisions and choices that you make and the food that you eat, and whatever you do, it’s obviously affected by this [EDCs] all the time and you don’t really notice it.”(FG2F)

Conversely, other participants believed EDCs to be a highly severe risk, especially with regards to the reproductive health of future generations of women, and the effects that this will have on the population:
“A high risk, absolutely high risk, because in the end, for women especially, if you’re endocrinal system doesn’t work you don’t get your period often, which means you don’t ovulate, which means eventually our population will be declining because we won’t be able to make children, and that’s because our systems don’t work, and it’s a big problem.”(FG2F)

#### 3.3.3. Perceived Control of EDC Exposure

Overall, participants did not feel they had much personal control over being exposed to EDCs, mentioning that these chemicals are so ubiquitous in modern life. Instead, it was agreed that the control is in the hands of the authorities and government, and a general sense of naivety was felt in having these beliefs:
“It’s a lottery; these things are beyond our control. Yes, maybe it’s a bit naive, but we assume that the authorities are taking care of us.”(FG4M)

Participants expected the government and authorities to protect them from such harmful chemicals and felt very little personal responsibility for their own exposure. There was speculation that the government and authorities had not sufficiently done their part to communicate the risks to the public. There was also a uniform feeling of disbelief that these chemicals were so present in modern life, and participants agreed that their perceived lack of control stemmed from an inability to avoid being exposed to these chemicals. A general attitude that this was the “price” people pay for choosing to live a certain lifestyle was apparent:
“It’s kind of just wrapped in with modern life, isn’t it? You know they’re there but. It’s very difficult to avoid them. I think a lot of people have just priced it in as the basis for the way they live.”(FG3M)

Participants felt EDC exposure was inevitable and thus individual protective behaviors may be futile:
“Well, some of these chemicals you cannot get rid of, even if you try to be careful.”(FG4M)

They were discouraged that there was very little they could personally do to alleviate their risk since these chemicals were already in the environment and difficult to avoid despite efforts to reduce their exposure:
“You can’t really do anything right without being exposed to some sort of bad thing.”(FG2F)

The unavoidable nature of EDCs left participants feeling despondent, especially after learning that BPA-free products were not necessarily a safer option. These BPA alternative products are typically labelled as “BPA-free.” However, due to their structural similarities with BPA, these alternative chemicals also show endocrine disrupting effects and adverse health effects, similar to BPA [29,30]. Participants bought BPA-free products thinking they had regained some control, but when they learned that BPA alternatives may be just as unsafe as BPA they were upset:
“I thought everyone was doing great, getting these non-disposable things, but there’s even more problems now.”(FG2F)

Every product has some amount of potential toxicity, and overall participants felt that if a person was to eliminate products on the basis that they were somewhat harmful, they wouldn’t be left with much else to consume:
“You can’t have a warning on all the products that potentially could cause harm…you’d walk into Tesco’s and everything would just be toxic, you know, if you go anywhere at all it’s going to be flashing at you, don’t drink this, don’t eat this.”(FG3M)

In order to regain any personal control over EDC exposure, there was a perception amongst participants that they would have to make drastic lifestyle changes. A recurring example that was given on how to regain control regarded the personal decision to have children. Participants felt that if a person was so worried about future generations being exposed, they could make the decision not to have children. Older participants who already had children or who were past the child bearing age proposed that a person could control their exposure by relocating to a less populated area or sustain themselves by living off of natural plants.
“I suppose it’s very difficult for us not to have them... Because of all the chemicals that are produced into the world… Unless you live in a wee hut somewhere and just eat from the natural plants.”(FG5F)

#### 3.3.4. Concern for Pre-Existing Health Conditions

All participants agreed that EDCs had the potential to be very dangerous for human health and had personal concerns about the issue. Participants who had pre-existing health conditions were especially concerned about the risks of EDCs.
“I’m very concerned about it being in the air, because I’ve problems with my sinuses that are quite bad and the idea that plastics could go into my sinuses and cause me bother is a bit horrifying, really.”(FG2F)

Female participants who were at the age of menopause were extremely worried about the presence of EDCs in soy products, since soy is often recommended to women going through menopause as a way to help alleviate negative symptoms. Although in small quantities soy can be helpful for women during menopause, these female participants were troubled and concerned that they had not been given information on EDCs from their doctors:
“I’m shocked at the cabbage and the sesame seeds and the soya beans, because, I mean, even the soya, you’re told when you were going through the menopause that soya was much better for you than ordinary milk.”(FG5F)

#### 3.3.5. Concern for Future Generations

Participants in the female focus groups were concerned for the health of their children and future generations while remaining indifferent towards their own exposure to EDCs. They wanted to make sure that their children would not be affected:
“Whatever about me, I want my children to have the best chance.”(FG2F)

Having children was the strongest motivator for them to take part in any risk reduction behaviors.
“Although I know about the chemicals, I wouldn’t really think that I was at risk, but if I was pregnant I would probably be more cautious or concerned about it, because like you said, breast milk and the placenta all transfer.”(FG2F)

Participants drew conclusions that hormonal contraception was a form of endocrine disruption and thus may affect fertility later in life. Although research has shown that female fertility is not negatively affected after the discontinuation of hormonal birth control [31], participants were concerned that birth control pills had endocrine disrupting effects and thus impacted the reproductive health of their daughters:
“My daughter… had to go and have tests and all that sort of thing done too as well, and it was all to do with her thyroid … it was just scary to think that her choice was not to have children younger, so she obviously was on the pill.”(FG6F)

### 3.4. Risk Alleviation Strategies

Participants wanted more communication from health authorities and the government. Several risk alleviating strategies were proposed by participants, that included educating themselves about the issues, changing aspects of their consumerism, or ignoring the risks altogether. A willingness to engage in risk reduction behaviors was observed, although there was a belief that the ubiquity of EDCs within modern life made it pointless to attempt to avoid them.

#### 3.4.1. The Role of Education

Participants highlighted the role of education in risk prevention and they valued being informed as it created a feeling of empowerment. They felt that by educating themselves about the issues, they could reduce their risk of being exposed to EDCs by being aware of which products are safe for consumption:
“It’s a matter of learning what not to be put into your body as best you can.”(FG5F)

Another educational tactic suggested was to learn how to correctly read product and food labels. In order to reduce their exposure to EDCs, participants wanted to learn what chemicals to avoid and teach themselves how to properly read the labels of their products with vigilance:
“Before you eat anything, you could check and see if any of those chemicals are inside it. You could check shampoo and things before you use it.”(FG1F)

High levels of perceived self-efficacy to engage in risk reduction behaviors were reported. However, participants still thought that they might become too difficult to implement, and that it would require their constant attention:
“If you think about them [the risk reduction guidelines], they look easy, but it would require constant vigilance, you would actually have to read every label to avoid things.”(FG3M)

#### 3.4.2. Implementing Lifestyle Changes

EDC risk could be alleviated by changing specific aspects of the participants’ consumerism, such as the products they use or the places they buy food. Participants suggested that they could reduce their exposure to chemicals in plastics by using reusable shopping bags, metal straws, and cardboard. A proposal was made about switching to organic, local produce, and shopping at farmers’ markets as opposed to large supermarkets:
“Go to farmers’ markets and look for fresh fruit and vegetables rather than going to bigger places, because they’re all packaged in plastic.”(FG5M)

Reverting back to traditional practices, such as home growing food and using reusable cloth diapers for babies was another tactic that was proposed by the participants:
“You’re better off growing fresh veggies, giving them a good wash, and doing your own baking rather than buying shop-bought stuff”(FG6F)

Although participants believed that they could reduce or alleviate the risks of EDCs by educating themselves or making lifestyle changes, there was a belief that the general public preferred to remain ignorant to the risks of EDCs. There were arguments that the younger generations would rather remain uninformed, but also arguments that older generations are too set in their ways and would not likely take on all of this new information. Participants discussed the ease of accessing new information about the risks of EDCs via the internet and media, but doubted whether people would actually implement the behavior changes within their daily lives.

## 4. Discussion

Using a qualitative design, this study examined the previously unexplored area of public perceptions of EDCs, their knowledge and awareness of EDCs and underlying themes of risk perceptions of EDCs.

In line with previous research [17,18,19,20], most participants in this study were not aware of EDCs. Those who were familiar with EDCs had either learned about them during an undergraduate science degree, were personally suffering from an endocrine related disorder, or had seen social media posts on EDCs from their cancer support group leader. The level of knowledge [32] and prior personal experience [33] with a risk are two factors known to help explain how risks are perceived by the general public. The lack of awareness found in this study indicates that there is a need for more information on EDCs to be included in educational and health systems. Additionally, the use of social media as a tool for spreading awareness could be useful in increasing knowledge, specifically when posted by someone who is familiar and trusted by the public.

Participants were much more aware of specific EDCs, namely pesticides and BPA. However, although participants were able to recall the names of specific EDCs, their knowledge about the chemicals was limited. Participants were not familiar with the specific EDCs due to their associations with disrupted endocrine functioning or their categorization as an EDC. Instead, it was evident that participants were familiar with these chemicals because of their widely known carcinogenic associations, environmental impacts, or other associated health risks outside of endocrine health. It may be difficult for the public to piece together their knowledge about specific chemicals to form comprehensive judgements about EDCs, as previous studies indicate that most individuals have difficulty parsing multiple sources of information together when making judgements and decisions [34]. The findings from this study support the position that communication of EDC risks should be broad and comprehensive in order to raise awareness of the endocrine disrupting effects of well-known chemicals.

The discussions also elicited some of the most common themes that were associated with participants’ risk perceptions of EDCs, including: similarity heuristics, perceived severity, perceived control, concern for pre-existing health conditions, and concern for future generations. Previous research suggests that the public tend to base their judgments on health and environmental risks from their cognitive heuristics [35,36]. Since many participants were learning about EDCs for the first time, they often relied on similarity heuristics, or “mental shortcuts”, to make judgments about EDCs based on similarities with other health risks [37]. The similarity heuristic is an efficient process that was used by participants to compare EDCs to health hazards, such as smoking cigarettes, or having high cholesterol, that they were more familiar with. This helped participants to use their knowledge about comparable hazards to assist the formation of opinions on the newly introduced subject of EDCs. In cases such as EDCs, where there is a great deal of uncertainty and lack of public knowledge, similarity heuristics may allow for the less-is-more effect [38] and serve as a fast mental reference for everyday decision making. However, participants in this study occasionally formed inaccurate opinions (e.g., with regard to the safety of organic products) so this process of risk comparison has the possibility of resulting in the spread of misinformation.

Perceived severity of EDCs varied greatly among participants. For some participants, EDC risks were not as severe or important as other risks, including Brexit, climate change, and the obesity epidemic. One reason as to why this may be the case, is that the media does not report about EDCs as frequently as the aforementioned risks that the participants deemed as being more important. In 2006, a large survey by the European Commission found that in comparison to obesity, alcohol consumption, smoking tobacco, and infectious diseases, the public were least likely to recall receiving information in the media about chemicals that are harmful to human health [39]. It is possible that the participants in the present study did not perceive the health risk of EDCs to be as high as other risks due to their absence within mainstream media. In addition to not being reported frequently by the media, the effects of EDCs are often chronic and cumulative throughout a lifetime. Although health issues are key concerns for society, the effects of EDCs are not visible or immediate. Thus, the participants did not view EDCs as immediately threatening, similar to other studies on chemical risk perception [40]. The gradual nature of EDCs may contribute to the participants’ perception that EDC risk was not quite as severe as other risks.

In contrast, for some participants, EDCs posed a severe risk to their own health and that of their children and future generations. This is in line with previous European research where women were more concerned about the effects of chemicals on health, compared to men [41]. In the present study, participants who believed EDCs pose a severe risk to health shared their concerns in relation to reproductive capabilities, pregnancy, and menopause.

Perceptions of control are essential for the adoption of preventative health behaviors and making important health decisions [42]. Participants felt that their exposure to EDCs was out of their control due to the ubiquity and unavoidable nature of EDCs. They believed the control to be in the hands of the authorities and government, a commonly held belief by the public with regard to their perceptions of chemical risks [39]. Links between perceived personal control and chemical risk perceptions have been identified [43], and judgements about perceived control are important as they often impact the public’s participation in risk reduction behaviors [44]. Future risk communications of EDCs should aim to better articulate the steps that the public may take to prevent their exposure to EDCs and re-establish a sense of perceived personal control.

Concerns about pre-existing health conditions and about the health of children were common among the participants. Findings from the present study indicate that having a pre-existing health condition contributed to increased risk perceptions of EDCs. This may be the case due to the high comorbidity between pre-existing health conditions and anxiety [45], as well as the fact that impaired health may make an individual more susceptible to EDC risks. Worries about children’s health or the health of future generations were indicated by the participants in this study, specifically in the female groups. Due to the small sample size and uneven number of male and female participants, we are unable to conclude that this was a female centric concern. In our particular study, the concern for future generations was not present in male focus groups, but this finding cannot be reported as a valid gender difference due to the format of this study. However, future research should aim to compare male and female perceptions of EDCs using larger and more balanced sample groups.

Throughout the focus group discussions, participants made numerous inaccurate judgements based on the information given to them on EDCs. Participants believed that buying organic products was safer than non-organic products and that EDCs would be on the ingredients list of products, which is not the case since many EDCs are contaminants not purposeful ingredients. While these misconceptions were easily rectified after the discussions, it touches on the likelihood that the information on EDCs may be misinterpreted. Specifically, the belief that organic products are safer is a common misperception among the public [46]. There is inconclusive scientific evidence to suggest that organic products are any safer compared to conventional farming practices when it comes to EDC contamination [47]. Consumption of organic products may reduce the exposure to pesticide residues, but EDCs are present in soil and irrigation water, and bioaccumulate in the food chain. Regardless of how organic products are grown, they are still susceptible to the uptake and bioaccumulation of EDCs, and participants were not aware of this.

Participants were corrected on these inaccuracies after the termination of the focus groups, but risk communication strategists should be cognisant of these misconceptions in the future. Participants claimed to get information about EDCs on social media from someone who they consider to be familiar and trustworthy. However, it is important to note that this type of person may not have the expertise to disseminate accurate information. In the future, the public must be educated about reliable and trusted information sources, such as government organizations, who strictly rely on scientific evidence and have no personal bias or motive to spread false information.

### Strengths and Limitations

This study offers a valuable insight into the general public’s awareness and risk perceptions of EDCs. Data saturation was achieved, gaining views of a varied sample from a range of locations and socioeconomic profiles. Nonetheless, the findings from this study are prone to selection bias, since the participants were volunteers and the recruitment method more so represented the views of a predominately urban population, with an under-representation of individuals from rural areas. Even though the call for participation was open for both males and females, the majority of respondents were females, common in qualitative research [48]. Alternative strategies and locations need to be considered by researchers who wish to recruit more male participants, which might offer further insight into the potential impact gender may have on EDC risk perception.

A cancer support group was used to assist in the recruitment of the two final focus groups. These participants may have been more severely impacted by EDCs, but in the current study there were no differences in awareness or risk perception identified in comparison with the other groups. Being a member of this support group did not add any new themes during data analysis. However, due to the design of this study, comparing the cancer support group with the other four groups is not possible due to the imbalanced sample sizes. Exploring the impact that a personal experience with cancer has on perceptions of EDCs would be a valuable aspect to consider studying in future research on risk perception.

Further investigation of public risk perceptions of EDCs is needed to inform efficient risk communication strategies. Unlike other health risk prevention campaigns, the ubiquity of EDCs presents a difficult task to risk communicators, and care must be taken in the presentation of information about EDCs so that they do not discourage the public from engaging in preventative behaviors. The results of this study may help EDC risk communicators in developing useful tactics to reduce exposure and provide assurance that efficient systems such as the EU Commission are in place to mitigate EDC risks. The provision of such information will be of interest to the public, given their concerns regarding EDC sources and the associated health effects. Although the current study provides valuable insights into public perceptions of EDCs, their sources, and associated health effects, the exploratory nature of the present study does not warrant generalizations about the general population. Therefore, further quantitative studies are necessary to identify the determinants of public risk perceptions associated with exposure to EDCs.

## 5. Conclusions

This was the first study to qualitatively explore public perceptions of EDCs in a general context. Public awareness of EDCs, their sources, and associated health effect was low. This may be due to the lack of attention EDCs get from the mainstream media, the absence of information being distributed by health professionals, and the lack of educational resources in schools. Risk perception of EDCs varied greatly among participants. Some of the most common factors that influenced participants’ risk perception of EDCs, included: similarity heuristics, perceived severity, perceived control, inevitability, pre-existing health conditions, and children’s health. Knowing these influential factors may help decision makers to gain insight into the public’s perspectives on EDC risk and improve the effectiveness of risk communication strategies.

It is proposed that public awareness campaigns by the media, health care practitioners, and schools are some approaches that should be considered to increase public awareness and knowledge of EDCs. Policies for raising awareness amongst the public are needed. These policies should emphasize social interventions, such as health education and risk communication programs. Specifically, the critical links between EDC exposure and reproductive health should be included in these programs. Future research should aim to gain a more comprehensive view by using quantitative methodology with a larger sample size to validate the findings from this study. The findings from this study should be used to aid the development of effective risk communication strategies and public health interventions to improve the protection of the public from EDC risks.

## Figures and Tables

**Table 1 ijerph-17-07778-t001:** Focus Group Demographics.

Group	Location	Gender	Age Mean/SD	Age Range	Number of Participants
FG1F	Belfast	Female	19 ± 0	19–19	3
FG2F	Belfast	Female	25.5 ± 4	21–42	8
FG3M	Belfast	Male	57.7 ± 4	52–65	6
FG4M	Belfast	Male	38.6 ± 8	25–58	7
FG5F	Derry	Female	53 ± 19	23–65	5
FG6F	Derry	Female	49 ± 11	40–60	5

**Table 2 ijerph-17-07778-t002:** Outline of Focus Group Topic Guide. EDC: endocrine disrupting chemicals.

Topic	Example Questions
Key questions on awareness of EDCs Question	Has anyone ever heard of “endocrine disrupting chemicals” before?
Key questions on awareness of Specific EDCs	Can you name any specific chemicals that are thought to cause endocrine disruption? How do you think they can affect health? Do you know if they are associated with any specific health effects/disorders?
Key questions on risk perceptions of EDCs	How do you feel now that you know where these EDCs are found? How do you feel now that you know about the health risks associated with EDCs?

**Table 3 ijerph-17-07778-t003:** Characteristics of focus group participants, *n* = 34.

Characteristics	*n* = 34	
Characteristics in Frequencies	*n*	%
Gender		
Male	13	38
Female	21	62
Highest Level of Education		
None	0	0
Primary School	1	<1
Secondary School	9	26
Additional Training (NVQ ^1^, BTEC ^2^, etc.)	5	15
Undergraduate University	15	44
Postgraduate University	4	12
Parent		
Yes	11	32
No	23	68

^1^ National Vocational Qualification (NVQ), ^2^ Business and Technology Education Council (BTEC).

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
