# Peer review of "Public Awareness and Risk Perceptions of Endocrine Disrupting Chemicals: A Qualitative Study"

_ijerph, 2020, doi:10.3390/ijerph17217778_

Round 1

Reviewer 1 Report

Overall, the information presented in the manuscript is interesting and provides insight into public perceptions on EDCs. However, there are some major concerns that must be addressed before this manuscript can be published. 

  • Lines 25, 48, 49, etc. - references state [e.g., 1]. I have no idea why e.g. is in the reference. Please fix this or clarify. 
  • Line 44 - appears that the word "and" is missing between "organisms" and "they"
  • Line 76 - should read "transparent with the general public". The word "the" is missing.
  • Section 2.2 - There is no mention that a cancer support group was used to assist with recruitment. This is a major concern because participants from this group may have skewed data given their own health conditions. Do you have any data as to whether the participants from this group had differing views from the others. This needs to be addressed in the manuscript with full transparency, and the data should be analyzed to determine if data was skewed in any way. The number of participants recruited from this group also needs to be identified.
  • Section 2.2- It is stated that the participants had no formal education with regard to chemicals from the food and/or environment, but later state that some participants were aware of EDCs from college science courses, which indicates that they do have some formal education on this topic. This needs to be better explained and resolved. 
  • Lines 140-147 and 165-178 - This section should be re-worded and not include the research team's initials. Using initials is very odd and doesn't add to the clarity. 
  • Lines 149-154 - The manuscript states that participants were given an information packet but it is unclear where the information in the packet came from and whether it was scientifically reputable sources. Moreover,  the lack of clarity makes me wonder if the packet could have influenced the results of this study. There needs to be more information and clarity on this presented in the manuscript. 
  • Table 2 - Please define NVQ and BTEC as I have no idea what these mean. Alternatively, you could just specifically state examples. 
  • Line 204 - Should read "The majority of..."
  • Line 245-247 - The manuscript states that pregnant women and teenagers are at most risk due to hormonal changes during these life stages, but there doesn't appear to be any discussion as to whether this statement is even scientifically valid or just a misconception. please address any of these instances through the manuscript. 
  • Results overall - Although the manuscript is intended to be qualitative and not quantitative, the authors commonly use quantitative words such as "many" and "some". This then makes the manuscript both qualitative and quantitative, and as such, you need to include the quantitative data as well. There is no way to accurately ascertain the legitimacy of the data presented without this data being included. It would be appropriate to add this as supplementary data. 

Additionally, the manuscript only offers one person's viewpoint on each result, which does not necessarily support the data that the authors present or the overall beliefs of the study group. there needs to be further clarification as to whether the quotes were representative of the entire (or majority) of the group, or you need to add additional data backing the results and conclusions. 

  • Line 333-336 - The manuscripts makes a claim that BPA-free products may be just as harmful as products with BPA. I have no idea where this claim comes from and whether this is even scientifically supported. Moreover, there is a huge concern that participants may have been misled or given inaccurate information and thus the data presented in the manuscript may be tainted. When claims about product safety are made, the claims need to be referenced using sound scientific sources. This needs to be corrected throughout the manuscript.
  • Section 3.3.5 - The manuscript again includes quantitative word usage by stating "It was common..." but doesn't provide the data. It also states that the concern was common among the female participants but doesn't include any information about  the male participants and data associated with the male participants. 
  • Lines 390-396 - It is unclear how birth control comes into play in this study, and I am unaware of birth control affecting the ability to produce offspring later in life. I am extremely concerned about this and there are no references given to even support this claim. Again, this leads to an overall concern that participants may have been given information that is not scientifically sound. This needs to be justified and referenced. 
  • Lines 411-417 - EDCs are not normally listed on labels because they are often a contaminant and not an ingredient. Moreover, the chemical may not yet have been identified as an EDC. Given this, it is quite questionable if reading labels would produce any positive outcomes in avoiding EDCs. Most consumers generally do not understand what most of the items on the ingredient list do, what they are, or how they work. 
  • Lines 425-438 - There are major misconceptions presented here including that organic, locally produced or farmer's market products are safer. Many EDCs are present in soil and irrigation water thus making any produce potentially susceptible to uptake and bioaccumulation of EDCs regardless of how they are grown or acquired by the general public. There is a continued suspicion that the participants may have been given information that is inaccurate and not scientifically sound. 
  • Line 445 - The authors make a statement about implementing behavior changes but these behavior changes are not really identified or discussed.
  • Line 453 - (see above) Again it is a huge concern that at least some participants may have previously received unreliable information from their cancer support group leader, and that their personal health issues may have skewed the data. This must be addressed.
  • Lines 456-458 - The manuscript discusses increasing knowledge by posting information from "someone who is familiar and trusted by the public". However, this type of person may not have the expertise to disseminate accurate information. 
  • Lines 471-483 - There is discussion about participants knowledge about comparable hazards assisting in the formation of opinions to drive everyday decision making. However, there is no discussion on whether these opinions formed are even accurate or whether this process results in further misinformation. This needs to be discussed. 
  • Lines 502-504 - Needs a reference for the statement that "women are likely to experience more hormonal changes throughout their lifetime..." because there doesn't appear to be a scientific basis that women's hormonal changes cause gender differences in exposure to EDCs. If this statement cannot be scientifically supported then it is pure conjecture and needs to be removed from the manuscript. 
  • Strengths and Limitations - There needs to be further discussion here that you in-part recruited from a cancer group who may have been more severely impacted by EDCs and thus have skewed the data. 
  • Overall- There remains a huge hole in the discussion. The discussion needs to be more in-depth regarding whether any misconceptions were formed by the participants during the study with regard to EDCs and whether any of their thoughts and opinions are scientifically valid. 

Reviewer 2 Report

ne 27 – There are also additional mechanisms of endocrine disruption, the authors could mention them;

Line 32-  “levels of EDCs concentrations” is redundant; The word level or the word concentration should be removed;

Line 48 – The sentence “Finally, transgenerational effects have also been observed, for example the effects of EDCs on male fertility” is not clear. Decreased male fertility is a transgenerational effect? I believe “transgenerational” refers to an effect observed in descendants;

Line 84 – Removing the words “all” would be more cautious, since any study will cover “all” these the EDs, “all” the effects etc;

Line 98 – Validation and recognition of focus groups methodology should be mentioned; I suggest using more references than reference 21;  

Line 129 – I believe that the information in these lines might not be necessary;

Table 1  - Considering the low number of participants (34), this study can be mentioned as a preliminary data;     

Line 149 – The information given previously to the participants should be provided in the paper, to prove that it did influenced the participant answers;

Table 3 – The relation between tables 1 and 3 is confusing and some information is repeated; The line concerning education “None” is not necessary; In additional training NVQ etc should be spelled out in the legend;

Line 194 – University degrees where DEs are lectured are generally specific – majority of the participants;  in line 201 “knowledge of the endocrine system” by many participants is also mentioned; how can the authors assure that these sample is not biased?; inversely in line 125 the authors mention “ had no prior formal education with regard to chemicals from food and/or the environment”; This should also be clarified;

Line 455 – Considering the used sample size, the authors can not use the term “general public” because it is not enough to extrapolate, as mentioned in line 542;

Line 498 – The indication of how many participants (or the %) considered EDCs as a severe risk would be more elucidative for the reader; The same for other issues along the discussion.

Round 2

Reviewer 1 Report

The manuscript has been greatly improved and I appreciate the authors' efforts.

Reviewer 2 Report

No comments regarding the new version.